# Fine-Grained Promote Learning for Face Anti-Spoofing

## ABSTRACT

There has been an increasing focus from researchers on Domain-Generalized (DG) Face Anti-Spoofing (FAS). However, existing methods aim to project a shared visual space through adversarial training, making it difficult to explore the space without losing semantic information. We investigate the inadequacies of DG that result from classifier overfitting to a significantly different domain distribution. To address this issue, we propose a novel Fine-Grained Prompt Learning (FGPL) based on Vision-Language Models (VLMs), such as CLIP, which can adaptively adjust weights for classifiers with text features to mitigate overfitting. Specifically, FGPL first motivates the prompts to learn content and domain semantic information by capturing Domain-Agnostic and Domain-Specific features. Furthermore, our prompts are designed to be category-generalized by diversifying the Domain-Specific prompts. Additionally, we design an Adaptive Convolutional Adapter (AC-adapter), which is implemented through an adaptive combination of Vanilla Convolution and Central Difference Convolution, to be inserted into the image encoder for quickly bridging the gap between general image recognition and FAS task. Extensive experiments demonstrate that the proposed FGPL is effective and outperforms state-of-the-art methods on several cross-domain datasets.

## CCS CONCEPTS

• **Computing methodologies → Computer vision**.

## KEYWORDS

Face Anti-Spoofing, CLIP, Prompt Learning, AC-adapter, Domain Generalization

## 1 INTRODUCTION

Face Recognition systems (FRs) are widely used in daily life due to their unique advantages, such as intuitive, real-time, and non-contact. However, these systems also face significant security risks as criminals may use Presentation Attacks (PAs), such as printed photographs, replayed videos, and 3D masks [28], to compromise the FRs to steal user information. Therefore, Face Anti-Spoofing (FAS) has become a crucial component in enhancing the security of FR systems against such malicious attacks with the growing importance in applications like face unlocking, face payments, and access control systems.

The existing Presentation Attack Detection (PAD) algorithms can usually be summarized as appearance-based and temporal-based

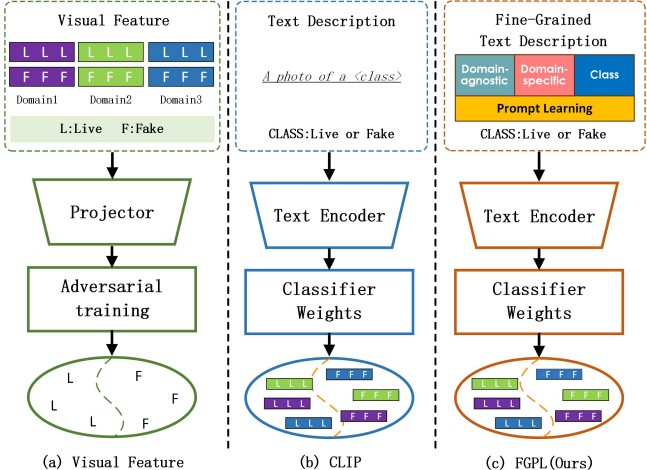

**Figure 1: (a): Existing methods only encode the visual image features to find the shared space through domain alignment. However, they miss semantic information due to direct editing of visual features through adversarial training. (b): The CLIP uses a fixed template text to describe "a photo of", which cannot accurately describe the subtle security features of FAS. (c): We employ Fine-Grained prompt learning (FGPL) based on the CLIP framework that treats domain generalizations-based FAS as high-quality textual feature learning and solves domain-generalized.**

methods. The goal of appearance-based techniques is to differentiate between genuine and artificial faces by utilizing various appearance cues, such as deep features [63], color textures [5, 6], and picture distortion cues [62]. On the contrary, a variety of temporal cues, including facial gestures [48, 49], rPPG [37, 38], and optical flow, may be extracted using temporal-based techniques. Shown as Fig. 1 (a), While these techniques have shown promising results in tests conducted within the same dataset, their effectiveness significantly declines when the training and testing data are obtained from different datasets. The chief reason for this deterioration is that these techniques, tailored solely to the training data, fail to bridge the gap between source and target domains, resulting in inadequate generalization.

Domain Generalization (DG) [58] is a commonly used technique in cross-domain contexts to address this issue by learning domain invariant representations. Recently, some DG-based algorithms [10, 16, 36] aim to improve the performance of models on unknown datasets. Great generalization performances have been achieved by works that make use of meta-learning [35, 75], while others make use of adversarial learning [29, 47, 61]. These techniques are dedicated to acquiring a generalized feature space, assuming that the extracted unknown faces can be mapped into the generalization space, and improving the model's generalization by

removing Domain-Specific (DS) information. However, attempting to generate a shared feature space for FAS through adversarial training may result in the following issues: (1) Significant disparities in the distribution of different data domains can easily lead to classifiers overfitting DS information. (2) Forcing the removal of DS information through adversarial training can lead to the loss of semantic information or the destruction of semantic structures in visual features.

For the first issue, our analysis shows that the leading cause is that the static weights of the classifier cannot dynamically adapt to the changing DS information. Thus, assigning different weights to samples from various domains during training can significantly mitigate the problem. For the second issue, direct editing of visual features via adversarial training or decoupling learning causes loss of semantic information and structural damage. By using an adapter to modulate visual features towards generalization instead of directly editing them, we can avoid this problem. Inspired by Vision-Language Models (VLMs), such as CLIP [43], which can perform zero-shot inference with a set of weight vectors by embedding the names or descriptions of the target dataset's classes as depicted in Fig. 1 (b), we can adaptively adjust weights for the classifier with text features. Furthermore, we design a lightweight adapter to transfer CLIP knowledge to FAS tasks, improving the model's generalization ability with minimal learnable parameters and avoiding the loss of feature information. Therefore, in this work, we treat DG-based FAS as a high-quality text feature learning procedure with an effective adapter.

To avoid time-consuming and performance-unstable prompt engineering for text feature learning, CoOp [73] models a prompt's context words with learnable vectors while putting the [CLASS] (i.e., the names or descriptions of the target dataset's classes) token in the end or middle position. Furthermore, CoCoOp [72] alleviates overfitting in the base classes by learning a lightweight Meta-Net to generate an input-conditional token (vector) for each image. However, they generally learn a set of prompts as inputs to the text encoder to generate text features and cannot selectively suppress domain-related information. In this work, we propose a novel Fine-Grained Prompt Learning (FGPL), which first motivates prompts to learn semantic information of both content and domain by capturing Domain-Agnostic (DA) and Domain-Specific (DS) features. As is shown in Fig. 1 (c), the model's generalization ability is improved by reducing the correlation between DA and DS prompts. Finally, the joint prompts are further designed to be category-generalized by diversifying the DS prompts. Additionally, we design an Adaptive Convolutional Adapter (AC-adapter), which is implemented through an adaptive combination of Vanilla Convolution and Central Difference Convolution, to be inserted into the image encoder for quickly bridging the gap between general image recognition and FAS task. To sum up, the main contributions of this paper are summarized as follows:

- We propose a new strategy called Fine-Grained Prompt Learning (FGPL), which enhances the model's generalization ability by reducing the correlation between Domain-Agnostic and Domain-Specific prompts.
- We use Domain-Specific context in the prompt and diversification of Domain-Specific prompts, further design of

category-generalized joint prompts. The ultimate implementation of adaptive adjustment of classifier weights with text features.
- We design a lightweight Adaptive Convolutional Adapter (AC-adapter) that adaptively combines the Vanilla Convolution and the Central Difference Convolution. It enables the rapid integration of general image recognition and FAS tasks.
- Extensive experiments show that the proposed FGPL is effective and outperforms the state-of-the-art methods on several cross-domain datasets.

## 2 RELATED WORK

### 2.1 Face Anti-Spoofing

During the initial phases, researchers have presented handcrafted feature-based presentation attack detection (PAD) methods [13, 31, 33, 40]. The majority of conventional algorithms are developed using manually crafted features that rely on abundant texture and temporal appearance cues. These cues include LBP [17, 39], HOG [21], SURF [31], SIFT [41], facial and head movements [3, 17], such as smiling and nodding, eye-blinking [26, 32, 40], gaze tracking [2, 4], and remote physiological signals, for example rPPG [21, 33]. While the aforementioned approaches have yielded noteworthy outcomes, their applicability is restricted to test data originating from the same domain. When the training and testing data are sourced from distinct domains, the performance of these methods significantly deteriorates.

### 2.2 Domain Generalization Methods

In order to cope with the identification of unseen domains, Domain Generalization (DG) becomes a more effective approach to address the unseen domain for FAS. The initial proposition by Shao et al. [47] involves the acquisition of a generalized feature space that is shared among many source domains through the utilization of a multi-adversarial discriminative domain generalization framework. Wang et al. [56] proposed a method for distinguishing between generic Facial Attribute Synthesis features from subject discriminative features, as well as domain-dependent features. Jia et al. [29] proposed a method for obtaining a discriminative and generalized feature space. The study [16] utilizes adversarial domain adaptation as a method to acquire knowledge in a shared embedding space. The paper [36] employs several domain discriminators to acquire knowledge in a comprehensive feature space. The authors in [16, 67] employ disentangled representation learning to separate the features associated with liveness for categorization purposes. To acquire comprehensive knowledge, numerous meta-learning-based approaches [11, 42, 50] have been developed and improved for the purpose of regular optimization. Despite the achievements in pursuing a mutually shared feature space, inherent constraints and drawbacks still need to be acknowledged and addressed, such as the loss of semantic information, which can further weaken category discrimination. Unlike existing strategies [29, 47, 56] for seeking common spaces in the field of FAS, we propose a novel approach using fine-grained prompt learning and taking advantage of the Vision-Language Models (VLMs) model to transform DG-FAS

problem into a high-quality text feature learning procedure with an effective adapter.

## 2.3 Vision-Language Models (VLMs)

Advances in computer vision [27, 70] have shown that the use of extensive pre-training with paired image-text data can be a viable alternative to achieving superior learning of visual representations without relying exclusively on natural language guidance. Since the Contrastive Language-Image Pretraining (CLIP) [43] was proposed, it has stimulated research into VLMs. Now, CLIP has been successfully applied in many fields, such as [22], image generation [45], visual question answering [1] and Domain Adaptation [19]. With further research [8], the construction of VLMs using independently pre-trained Large-Language Models (LLMs) and visual backbone models allows VLMs to understand both text and images with only a few parameters in the training phase. Inspired by the simultaneous comprehension of image-text in VLMs, we have delved further into the potential use of CLIP to enhance the FAS task.

## 2.4 Prompt Learning

The concept of prompt learning has roots in Natural Language Processing (NLP), a procedure that includes instructions at the beginning of the input sequence. Prompt learning aims to leverage these instructions to execute downstream tasks without necessitating fine-tuning. Many existing studies have used prompt learning [46, 51]. For instance, the CoOp [73] transforms pre-trained VLMs into data-efficient visual learners that outperform the original CLIP's hand-designed prompt learning templates. Subsequent research [14, 23, 60] has further advanced the development of CLIP, addressing a variety of aspects, particularly in terms of generalization capabilities. For example, by ensuring that prompts are relevant to the input image, the CoCoOp [72] demonstrates adaptation to new target domains, while the ProGrad model [76] achieves the same objective through gradient correction techniques. In recent times, there have been proposals such as the CLIP-adaptor [18] and the TIP-adaptor [68] that aim to enhance the transfer-ability of CLIP on downstream tasks by training a more efficient network. The DenseCLIP framework [44] utilizes the CLIP model to address dense prediction tasks through language-guided fine-tuning. Kg-CoOp [64] introduces a new knowledge-guided context optimization to enhance the generalization of learnable prompts to unseen classes. MaPLe [30] proposes multi-modal prompt learning for the Vision and Language branches. FLIP [52] is based on the Vision Transformer (ViT) visual model for fine-tuning, which aligns image representations with textual prompt Learning to achieve recognition of FAS tasks. However, these prompt learning described above are mostly applied in multi-category recognition tasks and require the category center to be learned using all the category names before the text encoding. In contrast, for the task of fine-grained FAS, simplistic and inflexible prompt learning may lead to overfitting of Domain-Specific information. To this end, we explored a Fine-Grained Promote Learning (FGPL) strategy, which dynamically adjusts the classifier weights through prompt learning, improving the generalization ability of FAS model while simultaneously allowing better application of VLMs to FAS tasks.

## 3 METHOD

### 3.1 Overall Framework

The architecture of the FGPL is shown in Fig. 2, which is built on CLIP. Unlike the standard CLIP [43], we design a fine-grained promote learning instead of fixed templates. Additionally, we add a lightweight AC-adapter to the image branch. In the following section, we explain our proposed approach in detail: Section 3.2 first reviews the CLIP approach and previous methods such as the CoOp. Then, we present our FGPL approach in section 3.3.

### 3.2 Revisiting CLIP

CLIP [43] comprises two encoders, one for image and one for text, which can jointly train an image encoder and a text encoder to predict the correct pairings of a set of <image, text> training examples. The contrastive learning objective aligns the image and text representation in the same feature space. CLIP encodes the input image $I \in \mathbb{R}^{H \times W \times 3}$ and the corresponding text description $t$ into a shared embedding space. A specific description is explained below.

**Image encoding:** The image encoder is responsible for converting an image into a feature vector, which can be implemented using either a ResNet [24] or a ViT [15]. Suppose the training set contains M samples, which denotes $S = \{I_i, T_i\}_{i=0}^{M-1}$, where $I_i \in \mathbb{R}^{H \times W \times 3}$ and $T_i$ is the textual description corresponding to the image $I_i$. $v(\cdot)$ is the visual encoder that encodes $I_i$ into a visual feature: $v_i = v(I_i)$, $v_i \in \mathbb{R}^d$, Where $d$ is the hidden dimension of the CLIP.

**Text encoding:** The text encoder accepts a series of word tokens as its input and generates a vectorized representation, which is implemented using a transformer [54]. The CLIP text encoder provides feature representations for text descriptions by tokenizing words and projecting them to word embeddings. $\tau(\cdot)$ is the textual encoder that encodes $T_i$ into textual feature: $t_i = \tau(T_i)$, $t_i \in \mathbb{R}^d$, Where $d$ is the hidden dimension of the CLIP.

**Zero-shot inference:** Using the classification problem as an example, CLIP can achieve zero-shot classification by correctly generating the text input. CLIP utilizes a fixed template prompt to form the text input, for example, "a photo of [$CLASS$]". The [$CLASS$] in the fixed template can be replaced with the actual class name. For fixed template $T_i' = \{A \text{ photo of } [CLASS_i]\}$, here $i$ is the $K - th$ categories, $i = \{0, 1, 2 \ldots, K\}$. Fixed samples are fed into the encoder to get text features $\{t_i'|t_i' = \mathcal{T}(T_i')\}_{i=0}^K$, therefore, the predicted probability of CLIP is as follows Eq. 1. where $\sin(\cdot)$ denotes the similarity calculation and $\tau$ is a temperature parameter.

$$p(y = i|I) = \frac{\exp(\sin(\boldsymbol{t_i'}, \boldsymbol{v})/\tau)}{\sum_{i=1}^K \exp(\sin(\boldsymbol{t_i'}, \boldsymbol{v})/\tau)} \tag{1}$$

However, CLIP fixed template prompts depend on manual settings and require word matching using a reserved validation set, which can be time-consuming. Therefore, CoOp [73] proposes prompt learning, which uses continuous $t_k$ representations to more accurately describe semantic features, as in Eq. 2. The $\mathbf{v}_{m_1}$ is a vector of the same dimension as the word embedding, and $m_1 \in \{1, 2, \ldots, M_1\}$.

$$\mathbf{t}_k = [\mathbf{v}]_1 [\mathbf{v}]_2 \ldots [\mathbf{v}]_{M_1} [\text{CLASS}]_k \tag{2}$$

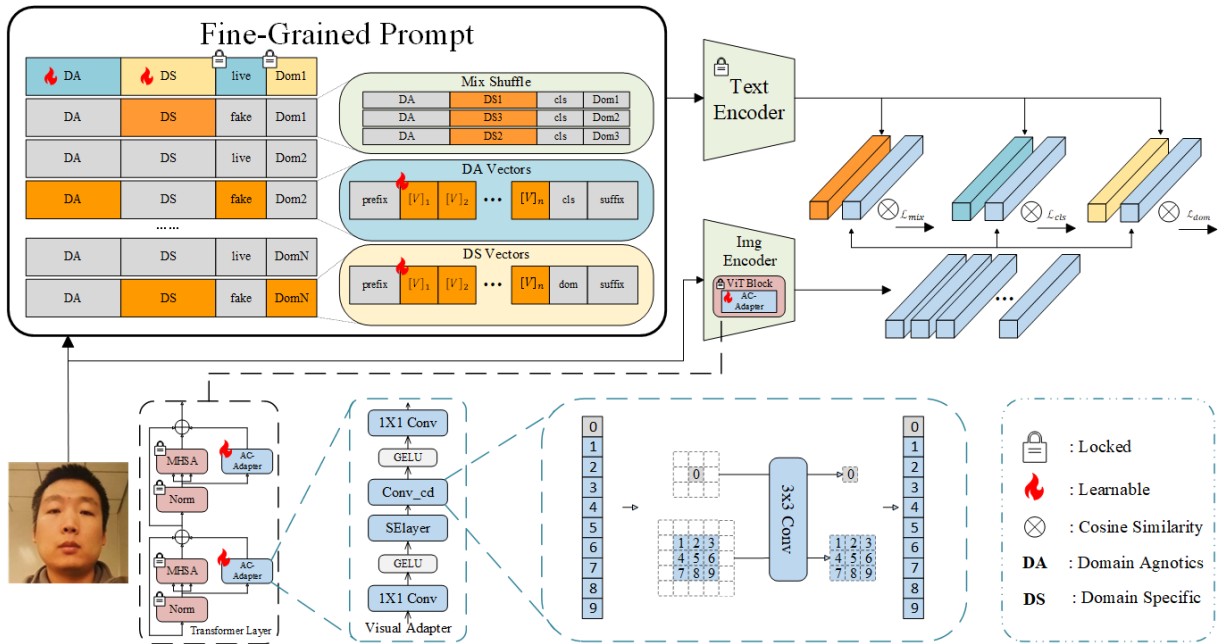

**Figure 2: Overview of the proposed FGPL framework. FGPL first motivates cue learning content and domain semantic information by capturing domain-independent and Domain-Specific features. Then, category-generalized common cues are further designed by diversifying the DS cues. The AC-adapter is implemented through an adaptive combination of Vanilla Convolution and Central Difference Convolution.**

## 3.3 Domain Generalization via Fine-Grain Prompt Learning

Due to the variations in Domain-Specific (DS) factors like illumination, background, and camera type, former methods using fixed text templates for domain descriptions often fail to capture complex domain information. This leads to misclassifications driven by spurious correlations among features from different domains. The existing CoOp [73] model, although innovative in employing continuous text sequences through prompt learning, does not integrate domain-specific information and lacks granularity in its prompts.

**Initialize Vectors.** To address these challenges, we introduce the concept of Fine-Grain Prompt Learning (FGPL).

FGPL incorporates learnable Domain-Agnostic (DA) and Domain-Specific (DS) contexts into the prompts to further guide the FAS task. The DA context is universal across all domains, while the DS context is tailored to individual domains, embedding pertinent domain information into the prompts. We define the DA and DS vectors as two learnable variables, i.e., $\mathcal{V}_{DA}$ and $\mathcal{V}_{DS}$, with a length of $L$, where $L = 16$ in our case. At first, $\mathcal{V}_{DA}$ is initialized in $\mathbb{R}^{16 \times 512}$, while $\mathcal{V}_{DS}$ is defined in $\mathbb{R}^{D \times 16 \times 512}$, containing domain information. Then, to further process and unify the prompts, we expand $\mathcal{V}_{DA} \in \mathbb{R}^{D \times cls \times 16 \times 512}$ to obtain the domain and class information. Also, $\mathcal{V}_{DS}$ has been expanded to include classification information. The architecture of $\mathcal{V}_{DA}$ and $\mathcal{V}_{DS}$ prompts are structured as Eq. 3 and 4:

$$\mathcal{V}_{DA} = [v_1, v_2, v_3, ..., v_L] \tag{3}$$

$$\mathcal{V}_{DS} = [v_1, v_2, v_3, ..., v_L] \tag{4}$$

Where $L$ is defined as 16 in our case.

**Construct Prompts.** We construct these vectors as Mix-Prompt $\mathcal{P}_{mix}$, Class-Prompt $\mathcal{P}_{cls}$, and Domain-Prompts $\mathcal{P}_{dom}$.

First, we construct a mix prompt $p_{mix}$ containing all information in the sequence $\{[sos][v_{DA}][v_{DS}][cls][dom][eos]\}$ and $[v_{DA}]$ and $[v_{DS}]$ in $p_{mix}$ is first filled with some $[x]$ placeholder. Also, the prefix, which is $\{[sos]\}$, is extracted for all three prompts, and the suffix $SFXmix = \{[cls][dom][eos]\}, SFXcls = \{[cls][eos]\}, SFXdom = \{[dom][eos]\}$ is extracted separately. Then, class prompt $p_{cls}$ and domain prompt $p_{dom}$ is structured by combining the prefix and suffix with $v_{DA}$ and $v_{DS}$ separately, where $v_{DA}$ and $v_{DS}$ here refer to the $[v_{DA}]$ and $[v_{DS}]$ in the mix prompt $p_{mix}$. Then, three prompts $p_{mix}, p_{cls}$, and $p_{dom}$ are tokenized for text feature extraction and further fed to the CLIP's embedding layer $Emb$ to generate each's embeddings.

Second, in order to obtain the final prompts for the classification, we adopt the prompt in the first stage and replace the $[v_{DA}]$ and $[v_{DS}]$ parts with the learnable DA and DS vectors. The structure of the Prompts is shown as follows:

$$\mathcal{P}_{mix} = \{[PFX_{mix}][\mathcal{V}_{DA}][\mathcal{V}_{DS}][SFX_{mix}]\} \tag{5}$$

$$\mathcal{P}_{cls} = \{[PFX_{cls}][\mathcal{V}_{DA}][SFX_{cls}]\} \tag{6}$$

$$\mathcal{P}_{dom} = \{[PFX_{dom}][\mathcal{V}_{DS}][SFX_{dom}]\} \tag{7}$$

Where $PFX$ refers to the prefix, and $SFX$ refers to the suffix of each prompt. $V_{DA}$ and $V_{DS}$ here represent the learnable DA and DS vectors.

For the Mix-Prompt $\mathcal{P}_{mix}$, we randomly shuffle the $[\mathcal{V}_{DS}]$ related to each domain, which makes the model less concentrated

on the domain-specific information. For Class-Prompt $\mathcal{P}_{cls}$ and Domain-Prompt $\mathcal{P}_{dom}$, we insert learnable DA and DS vectors, respectively, to focus on different aspects of the image.

Finally, the CLIP's text encoder further processes three prompts, with each tokenized prompt in the first stage, to obtain the text features of the images and calculate the cosine similarity of the image and text features. This refined approach to prompt learning improves the adaptability and effectiveness of face anti-spoofing models across varied domains. It sets a new standard for integrating domain-specific information into deep learning frameworks.

## 3.4 Adaptive Convolutional Adapter

The pivotal component of the FAS task is the precise identification and analysis of information that discriminates between live and fake representations. Initially, processing visual features through Vanilla Convolution can lead to semantic information loss and structural degradation. Furthermore, utilizing the image encoding capabilities of CLIP may hinder the learning of fine-grained details critical for FAS. This paper introduces a lightweight visual adapter, Adaptive Convolutional Adapter (AC-Adapter), which integrates into CLIP's visual encoding framework to address these issues. This adapter innovatively merges Vanilla Convolution with central difference convolution [66]. By incorporating the latter, the adapter enhances its capability to detect subtle, deceptive patterns by effectively combining luminance and gradient data, thus improving its effectiveness in distinguishing authentic from spoofed images. The adapter module is placed in parallel with the Multi-Head Self-Attention (MHSA) and Multi-Layer Perceptron (MLP) block, given an input $I \in \mathbb{R}^{N \times d}$:

$$I' \leftarrow I + \text{MHSA/MLP}(LN(I)) + v \cdot adapter(LN(I)) \quad (8)$$

As shown in the bottom of Fig. 2, the visual adapter module consists of four convolution layers: SElayer, which implements the Squeeze-and-Excitation (SE) procedure to adjust features across channels dynamically, is employed to enhance the network's capability for feature representation; Central Difference Convolution layer to compute the difference output; 1×1 standard convolution layer and 1×1 linear convolution to perform dimension change operations.

At first, the input $I \in \mathbb{R}^{N \times d}$, where N represents the length of the sequence and d represents the feature level (768), is reconstructed and downed to $dim = 8$ dimensions and further add non-linear features using GELU. Then, the last procedure's output $I^{dim}$ is fed to the SElayer, which contains two stages: Squeeze and Excitation. During the squeeze stage, $I^{dim} \in \mathbb{R}^{B \times dim \times 14 \times 14}$, where B and N represent the batch size and length of sequence respectively, are compressed through an adaptive average pooling operation, which outputs the global average for each channel, i.e., $I_{SE}^{dim} \in \mathbb{R}^{B \times dim}$. This procedure aggregates spatial information, compressing each channel's features into a single numerical value. Then, the SElayer, through a fully connected layer, reduces the dimensions of channels, followed by the introduction of non-linearity through a ReLU activation function, which also aids in mitigating the issue of vanishing gradients. Subsequently, the original dimensions are restored through another fully connected layer. Finally, a Sigmoid activation function outputs weights ranging from 0 to 1. These weights are

utilized to scale the features in each channel of the original feature map, completing the "Excitation" phase and dynamically adjusting the inter-channel feature responses.

Furthermore, the weights $W_{SE}^{dim} \in \mathbb{R}^{B \times dim \times 14 \times 14}$ of the SElayer outputs are passed into the Central Difference Convolution layer along with the production of a Vanilla Convolutional layer. The Central Difference Convolution layer first calculates the sum of the Vanilla Convolution weights across the last two dimensions to form a new kernel difference for computing a new convolution $I_{cd}^{dim} \in \mathbb{R}^{B \times N \times 14 \times 14}$. Then, by multiplying the output weights from the SElayer, the central difference convolution of the input is obtained, which is less than that which samples the local receptive field region $\mathbb{R}$ and will simultaneously capture the central gradient along with it. After the processing with this layer, the difference in results from the Vanilla Convolutional layer and itself has been calculated to extract differentiated features, shown as Eq. 9:

$$I'^{dim} = conv\_cd(I) - conv(I) \quad (9)$$

Where $I'^{dim}$ denotes the total output, $conv\_cd(I)$ represents the output of the Central Difference Convolution layer, which is the combination of the feature $I_{cd}^{dim}$ and the weight $W_{SE}^{dim}$.

Besides, the class token is obtained, convolved, and combined with the $I'^{dim}$ feature. Finally, the feature dimensions are restored to their original size of $\mathbb{R}^{N \times B \times 768}$.

In the FAS task, capturing semantic information at the intensity level and detailed information at the gradient level is essential for differentiating between live and spoofed faces. Therefore, employing such a hybrid approach that adaptively combines Vanilla and Central Difference Convolution is critical. The AC-adapter's forward propagation process includes dimensionality reduction via linear convolution, integration of SElayer parameters to facilitate both vanilla and center-difference convolution transformations and utilization of center-difference convolution subtracted from Vanilla Convolution to enhance the detection of forgery-specific features. Also, obtained from the adapters' structure, all trainable paths contain adapter modules after freezing the MHSA/MLP so that the whole fine-tuning process can benefit from the visual bias inherent to CNNs. Additionally, the Central Difference Convolution layer is implemented by transforming the receptive field from the original 3x3 neighborhood into sub-neighborhoods corresponding to horizontal-vertical or diagonal directions, making the visual adapter more focused on capturing Anti-Spoofing detail information.

## 3.5 Loss Function.

The image and text encoders are unfrozen during training, while fine-grained prompt learning and AC-adapter are added. After the encoders, three fine-grained text features and the image feature are then calculated for similarity and the predictions of the input image using Eq. 1. We utilize the standard CrossEntropy loss during training to obtain the fine-grained loss upon three kinds of features and then sum it as the total loss as shown in Eq. 10:

$$\mathcal{L}_{total} = \mathcal{L}_{cls} + \mathcal{L}_{dom} + \mathcal{L}_{mix} \quad (10)$$

Specifically, the Class-Loss $\mathcal{L}_{cls}$ utilizes the Class-Prompt $\mathcal{P}_{cls}$, the Domain-Loss $\mathcal{L}_{dom}$ uses the Domain-Prompt $\mathcal{P}_{dom}$ and the

**Table 1: The results (%) of Protocol 1 on MSU-MFSD (M), CASIA-FASD (C), ReplayAttack (I), and OULU-NPU (O) datasets.**

| Method | OCI→M | | OMI→C | | OCM→I | | ICM→O | | avg. |
|---|---|---|---|---|---|---|---|---|---|
| | HTER↓ | AUC | HTER | AUC | HTER | AUC | HTER | AUC | |
| MADDG [47] | 17.69 | 88.06 | 24.50 | 84.51 | 22.19 | 84.99 | 27.98 | 80.02 | 23.09 |
| DR-MD-Net [57] | 17.02 | 90.10 | 19.68 | 87.43 | 20.87 | 86.72 | 25.02 | 81.47 | 20.64 |
| RFMeta [50] | 13.89 | 93.98 | 20.27 | 88.16 | 17.30 | 90.48 | 16.45 | 91.16 | 16.97 |
| NAS-FAS [65] | 19.53 | 88.63 | 16.54 | 90.18 | 14.51 | 93.84 | 13.80 | 93.43 | 16.09 |
| $D^2AM$ [10] | 12.70 | 95.66 | 20.98 | 85.58 | 15.43 | 91.22 | 15.27 | 90.87 | 16.09 |
| SDA [59] | 15.40 | 91.80 | 24.50 | 84.40 | 15.60 | 90.10 | 23.10 | 84.30 | 19.65 |
| DRDG [35] | 12.43 | 95.81 | 19.05 | 88.79 | 15.56 | 91.79 | 15.63 | 91.75 | 15.66 |
| ANRL [36] | 10.83 | 96.75 | 17.83 | 89.26 | 16.03 | 91.04 | 15.67 | 91.90 | 15.09 |
| SSDG-R [29] | 7.38 | 97.17 | 10.44 | 95.94 | 11.71 | 96.59 | 15.61 | 91.54 | 11.28 |
| SSAN-R [61] | 6.67 | 98.75 | 10.00 | 96.67 | 8.88 | 96.79 | 13.72 | 93.63 | 9.81 |
| PatchNet [55] | 7.10 | 98.46 | 11.33 | 94.58 | 13.40 | 95.67 | 11.82 | 95.07 | 10.91 |
| SA-FAS [53] | 5.95 | 96.55 | 8.78 | 95.37 | 6.58 | 97.54 | 10.00 | 96.23 | 7.82 |
| IADG [74] | 5.41 | 98.19 | 8.70 | 96.44 | 10.62 | 94.50 | 8.86 | 97.14 | 8.39 |
| CLIP-V [43] | 4.29 | 98.76 | 70.00 | 5.00 | 98.89 | 76.33 | 7.14 | 97.92 | 74.29 |
| CLIP [43] | 4.04 | 99.13 | 5.00 | 98.89 | 5.57 | 98.45 | 6.09 | 98.12 | 5.17 |
| CoOp [73] | 4.29 | 98.76 | 2.11 | 98.55 | 6.07 | 97.52 | 4.60 | 98.78 | 4.51 |
| CoCoOp [72] | 4.05 | 99.18 | 4.77 | 98.15 | 9.21 | 97.39 | 6.80 | 97.27 | 6.21 |
| **FGPL (Ours)** | **2.86** | **98.12** | **3.89** | **98.19** | **3.50** | **99.54** | **1.77** | **99.23** | **3.01** |

Mix-Loss adpots the Mix-Prompt $\mathcal{P}_{mix}$ which shuffles the $[\mathcal{V}_{DS}]$ component.

## 4 EXPERIMENTS

### 4.1 Experimental Setup

**Datasets and Protocols:** We used two protocols to assess generalizability. For Protocol 1, we used benchmark datasets that are publicly available to the FAS academic community, MUS-MFSD (M) [62], CASIA-FAD (C) [71], Idiap Replay-Attack (I) [12] and OULU-NPU (O) [7]. The four datasets differ for various reasons, concerning material, lighting, background, and resolution differences. As a result, there is a significant bias in the domain between these datasets. Following the established testing rules, each dataset is treated as a domain, and the other three source domains are combined. The leave-one-out test protocol is applied to assess the cross-domain generalizability of the method. For example, OCI→M, which uses OULU-NPU, CASIA-FAD, and Idiap Replay attacks as training protocols, was tested on MSU-MFSD. For Protocol 2, we use the large-scale FAS datasets CASIASRF (S) [69], CASIA CeFA (C) [34], and WMCA (W) [20], which are FAS proprietary datasets, as they encompass a broader range of topics, various types of attacks, and diverse sampling environments.

**Evaluation Metrics:** In line with the evaluation principles, we used three metrics, HTER, AUC, and TPR, to assess the model's performance. (1) HTER is a measure of the false rejection and false acceptance error rates, and its value is taken as the average of the false rejection rate (FRR) and the false acceptance rate (FAR). (2) AUC measures the algorithm's overall performance, and its value represents the area under the ROC curve. This metric is used to evaluate the performance of the classifier. (3) TPR (True Positive

Rate) measures the algorithm's accuracy in recognizing spoofed samples. It can select the appropriate threshold based on the specific application requirements.

**Implementation Details**: For the FAS task, we use a pre-trained CLIP [43] model with ViT-B/16 [15] as the image encoder. We keep the parameters in the encoder unchanged, resize the image to 224×224 with a batch size of 50, and train all models for 500 epochs. We use the Adam optimizer and set the learning rate to 1e-6 for training. Additionally, we set the length of the DA token M1 and the DS token M2 to 16 during the initialization of prompt learning.

### 4.2 Cross-domain FAS Performance

In Protocol 1, we present the results of the state-of-the-art (SOTA) approach and our improved results using the CLIP model. The following conclusions can be drawn from Tab. 1. FAS domain generalization can be effectively improved by using multiple variants of Vision Transformer (ViT). From the HTER metrics, it can be seen that the mean HTER values of CLIP [43] (5.17%), CoOp [73] (4.51%), and CoCoOp (6.21%) are significantly higher than the MADDG [47] (23.09%), $D^2AM$ [10] (16.09%), IADG [74] (8.39%), SA-FAS [53] (7.82%). The latter searches the commonality space through adversarial training, so the classifier weights do not adapt to dynamically changing domain-specific information. However, former methods conveying natural language semantic information have much more satisfying results. Such methods are well-established [23, 43, 72] that introducing natural language semantics enriches the visual representation and improves its generalization, enabling models to understand and reason about visual content more comprehensively and accurately. The latest research FLIP [52] also utilizes image-text. Still, it uses a fixed text template, which is sub-optimal for FAS tasks and does not allow for dynamic adjustment of classifier weights.

| Method | CS→W | | | SW→C | | | CW→S | | | avg. |
|---|---|---|---|---|---|---|---|---|---|---|
| | HTER↓ | AUC | TPR@FPR=1% | HTER | AUC | TPR@FPR=1% | HTER | AUC | TPR@FPR=1% | HTER |
| ViT [25] | 21.04 | 89.12 | 30.09 | 17.12 | 89.05 | 22.71 | 17.16 | 90.25 | 30.23 | 18.44 |
| CLIP-V [43] | 20.00 | 87.72 | 16.44 | 17.67 | 89.67 | 20.70 | 8.32 | 97.23 | 57.28 | 15.33 |
| CLIP [43] | 20.00 | 87.72 | 16.44 | 17.67 | 89.67 | 20.70 | 8.37 | 97.47 | 64.05 | 15.34 |
| **FGPL (Ours)** | **14.05** | **92.65** | **33.33** | **19.00** | **88.53** | **13.33** | **11.00** | **94.72** | **34.00** | **14.68** |

Table 2: The results (%) of Protocol 2 on CASIA-SURF (S), CASIA-SURF CeFA (C), and WMCA (W) datasets.

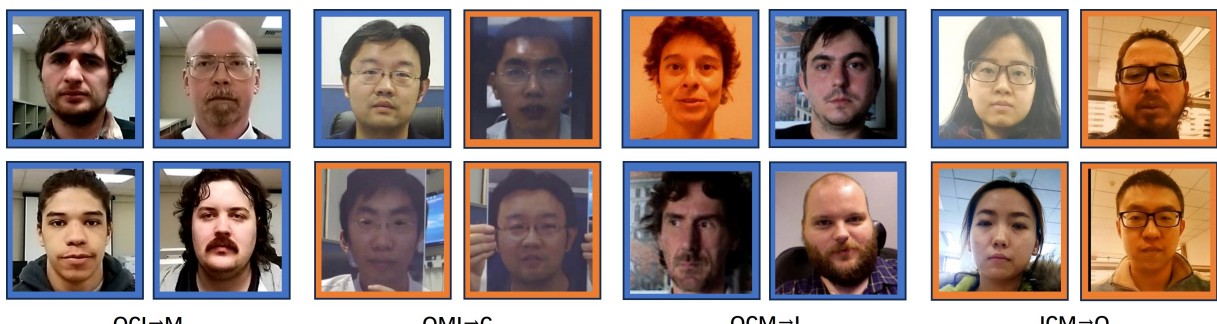

OCI→M  OMI→C  OCM→I  ICM→O

Figure 3: Example of mis-classificate in Protocol 1. Blue boxes indicate live faces that were misclassified as spoofed. Orange boxes indicate faces that have been misclassified as live.

Compared to the other SOTA methods, our approach significantly outperforms all baselines in terms of the mean value of HTER and most of the DG tasks.

In Protocol 2, The results of the different methods are presented in Tab. 2. Compared to ViT [15] and CLIP-V, which only involve image-side processing, and CLIP [43], which only uses fixed templates, the HTER of our FGPL is significantly improved. Unlike the baseline CLIP, FGPL incorporates fine-grained prompt learning and lightweight AC-adapters. Among them, constructing three-stage fine-grained textual cues achieves the precise localization of DA and DS information in visual features, effectively mitigating the overfitting problem of classifiers to domain-relevant information. On the other hand, the AC-adapter captures detailed deception features using an adaptive combination of Vanilla Convolutional and Central Difference Convolution. Compared to CLIP, CLIP-V, the HTER of FGPL is reduced by 0.66% on average on the MSU-MFSD (M) [62], Idiap Replay Attack (I) [12] and OULU-NPU (O) [7] datasets.

## 4.3 Ablation Study

To further validate the innovativeness of our approach, we first removed our FGPL's AC-adapter and replaced it with the original visual encoder in CoOp. Additionally, we replaced our Fine-Grain Prompt Learning with the original text prompts in CoOp. Through these experiments, we can simultaneously and collaboratively verify the effectiveness of our FGPL and AC-adapter. Shown as Tab. 3, the HTER score increased from 3.50% to 4.86%, and the TPR score decreased from 87.14% to 65.00% by removing the AC-adapter. Moreover, by stipping our FGPL, the HTER increased 3.93%. The results

show that the two modules we designed complement each other and are indispensable.

Table 3: Ablation of the structures for FGPL and AC-adapter on OCM→I.

| Method | HTER(%) | AUC(%) | TPR(%)@FPR=1% |
|---|---|---|---|
| CoOp | 6.07 | 97.52 | 70.00 |
| CoOp-FGPL | 4.86 | 98.71 | 65.00 |
| CoOp-AC adapter | 7.43 | 97.76 | 80.71 |
| **FGPL (Ours)** | **3.50** | **99.54** | **87.14** |

Where CoOp represents the original CoOp model without adjustments, CoOp-FGPL denotes the original CoOp's image encoder with FGPL, and CoOp-AC adapter means the CoOp model with our AC-adapter, not including the FGPL.

**Effectiveness of FGPL:** Shown in Tab. 3, we conducted the OCM→I experiment to confirm the impact of our FGPL. On the one hand, by removing the FGPL, which refines the prompt learning procedure to reduce the notice of DS information, the CoOp-AC adapter's HTER score increases 3.93% and the TPR drops 6.97%. Moreover, its HTER score even increases by 1.36% compared to the original CoOp model. On the other hand, by adding our FGPL into the standard CoOp model, the HTER improves by 1.21%. The results significantly demonstrate the importance of our FGPL. Even if we use an AC-adapter in isolation, allowing us to extract more semantic information from images; however, the absence of textual cues to supervise the training process can lead the model to focus on DS information overly, ultimately resulting in decreased performance.

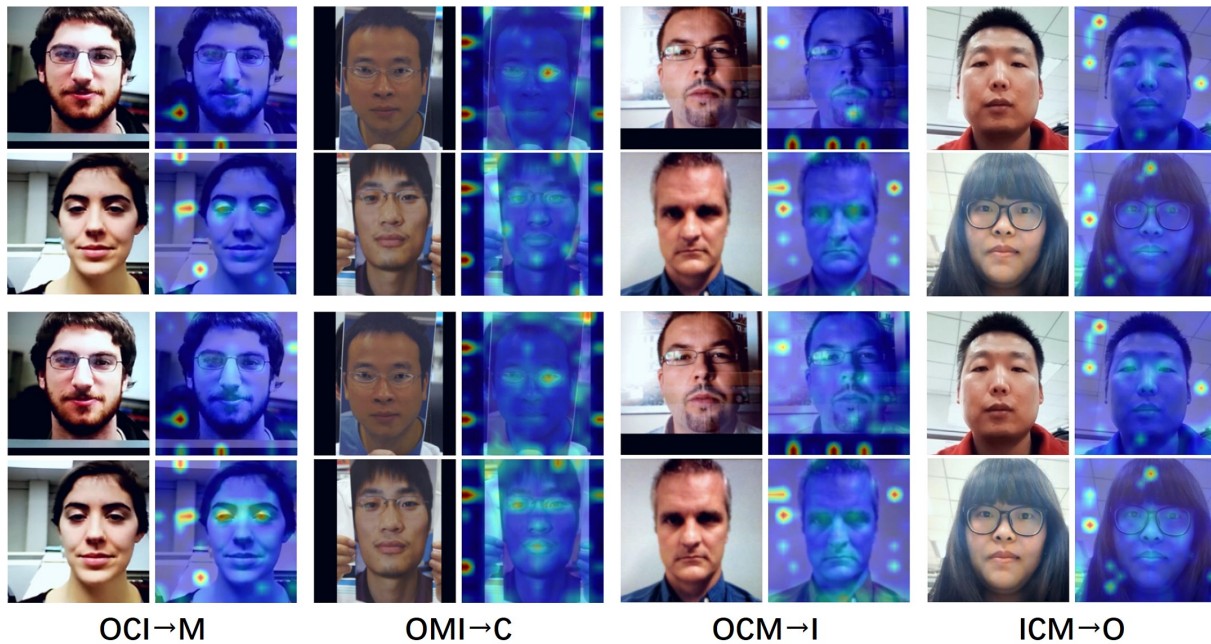

**Figure 4: Attention maps in Protocol 1. The first and second lines show the results using the baseline CLIP, while the third and fourth lines display the results of the FGPL.**

**Effectiveness of AC-adapter:** In the OCM→I experiments as Tab. 3, from one view, by removing the AC-adapter from our FGPL method, the HTER performance drops 1.36% and the TPR decreases 22.14%. From another view, by solely utilizing the AC-adapter based on the CoOp model, the TPR score increases 10.71%. Interestingly, the HTER score performance worsened if we used the AC-adapter without guidance from FGPL. This is because more fine-grained image features are dug, and DS information is also part of them. Without our FGPL DG method, the model can overfit the DS information and reduce the model's generalization performance. This verifies that our AC-adapter indeed captures more specific features and also restates the importance of using both modules as a whole.

## 4.4 Visualization

**Mis-classified images:** In Fig. 3, we present examples of mis-classified images in Protocol 1. It is easy to notice that none of the fake samples were misclassified in OCI→M and OCM→I. This can be attributed to the fact that the attack types in the training dataset contain the attack types in the test dataset. However, in OMI→C, approximately 6% of the samples are misclassified as fake faces. This issue may be attributed to the high resolution of the samples trained in OULU. Still, the CASIA test dataset samples do not have enough resolution, have low light conditions, or are overly bright. In contrast, the probability of misclassifying a live face as a fake face is extremely low at 2.1% in ICM→O. After all, the four benchmark datasets, OULU-NPU, are the higher resolution database. When the training samples are of low resolution, this can easily lead to recognizing a higher-resolution fake face as a live face in the testing

phase. Similarly, the same analytical conclusions apply to Protocol 2.

**Attention map:** To further demonstrate the benefits of FGPL, we used [9] to generate visual attention maps for the FGPL model on the deception samples in Protocol 1 and Protocol 2, respectively. As observed in Fig. 4, the dataset in Protocol 1 is primarily affected by printing and replay attacks. The baseline CLIP emphasizes untrustworthy spoofing cues like paper texture, edges, and ripples, resulting in the misclassification of samples. In contrast, our FGPL adjusts the model's focus on more subtle fake characteristics and can efficiently locate the fake patterns in each fake domain to make correct classification decisions. Likewise, the same analytical conclusions are applicable to Protocol 2.

## 5 CONCLUSION

In this paper, we consider DG-Based FAS as high-quality textual feature learning and effective adaptor design that improves the model's generalization capability with minimal loss of learnable parameters and feature information. FGPL proposes an effective framework for fine-grained prompt tuning. A refined prompt learner is used to optimize prompts to adjust classifier weights dynamically. A light Adaptive Convolutional Adapter (AC-adapter) quickly bridges the gap between general image recognition and FAS tasks. We have shown that visual language modules learned using pre-trained visual language models (e.g., CLIP) have excellent generalization capabilities in FAS tasks compared to models that only use multiple variants of ViT. Future work could explore more diverse prompt learning in conjunction with FAS image features to further improve the effectiveness of text monitoring in on-the-fly adaptation.

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
