# OpenReview forum: "Fine-Grained Prompt Learning for Face Anti-Spoofing"
_acmmm.org/ACMMM/2024/Conference — MM2024 Oral_

### Official Review · Reviewer_KEUN · 2024-04-28

**Rating:** 6
**Confidence:** 4

**Summary:**

This paper focuses on studying the domain-generalized Face Anti-Spoofing (FAS) task and proposes a novel Fine-Grained Prompt Learning (FGPL) method. Specifically, FGPL first motivates the prompt to learn content and domain semantic information by capturing both domain-agnostic and domain-specific features. Furthermore, the prompts are designed to be category-generalized by diversifying the domain-specific prompts. An Adaptive Convolutional Adapter (AC-adapter) is inserted into the image encoder to bridge the gap between general image recognition and the FAS task.

**Strengths:**

1. This paper presents a novel idea based on recent advanced Vision-Language Models (VLMs) to solve the domain-generalized FAS task. Based on CLIP, this paper mitigates the classifier's overfitting to the domain correlation information with the help of fine-grained text features, which is a solution that can directly hit the pain point. The FGPL proposed in this paper is a technique worthy of reference and further in-depth study by researchers.
2. Distinguishing from previous clip-based studies, the FGPL contains innovative modules such as fine-grained prompt learning and an Adaptive Convolutional Adapter. Among the former, the three-stage fine construction prompt accurately localizes domain-agnostic and domain-specific information in the visual features. The latter utilizes an adaptive combination of two convolutions to successfully transfer CLIP knowledge to the FAS task, enhancing the model's generalization with minimal learnable parameters and preventing the loss of crucial spoofing features. Therefore, this paper presents an innovative and valuable technique to solve the FAS domain generalization challenge by incorporating improved CLIP.
3. The paper is well-written and easy to understand. The insight about domain-generalized problems is interesting and well described, which helps to understand the proposed techniques.
4. Experimental results show the effectiveness of the proposed method.

**Limitations:**

1. The font size of the headings in the visible sections of the experimental part is inconsistent, and the authors should pay closer attention to minor layout details.
2. The literature could include more articles on Domain-Generalized (DG) Face Anti-Spoofing (FAS) research for 2023 and 2024.
3. The implementation code of the paper's method is not open-source.

**Suitability:**

3

---

### Official Review · Reviewer_K21f · 2024-05-16

**Rating:** 6
**Confidence:** 4

**Summary:**

This paper proposes a Fine-Grained Prompt Learning (FGPL) based on Vision-Language Models, such as CLIP, to solve the Domain Generalization in Face Anti-Spoofing area. The authors proposed a Fine-Grained Prompts which has two learnable vectors to capture Domain-Agnostic and Domain-Specific features. Furthermore, two learnable vectors are combined with the image feature separately. Concerning the image branch, the authors add an AC-adapter to dig the image features more deeply. The authors also conducted several DG experiments to verify the results.

**Strengths:**

1.	This paper considers the DG into a combined task of vision-language tasks, using both features to dynamically adjust the classifier’s weight. Although this paper addresses the DG problem in an ordinary motivation, but the utilization of Vision-Language multi-modal approaches is quite unique and novel. The learnable text vectors can learn the more detailed semantic information and guide the classifier more convincingly.
2.	The experiment of the proposed method is promising through the table showed in the paper. Two kinds of wildly used DG datasets have been adopted to verify the efficiency of the method.
3.	The loss function the authors had designed take all of three loss functions concerning the Domain-Agnostic and Domain-Specific features and the overall feature into the model’s training objectives, which concentrate on both global and local discriminative features.
4.	The AC-adapter utilizes the central difference convolution blocks that can help encoder capture semantic information at the intensity level and detailed information at the gradient level, which is essential for differentiating between live and spoofed faces.

**Limitations:**

1.	The written style and logic of the paper’s content is a little bit fuzzy that may lead readers to misunderstand what the authors want to tell.
2.	In the method part, although authors tried to use several formulas and symbols to describe their methodology more specifically, they should more concentrate on the meanings and the basic mechanisms which each vector can tell, not only what the vector is.
3.	The experiments of the proposed methods though promising compared to the baselines, it can also be compared to other VLMs-based methods, such as FLIP, which can have more convincing results.

**Suitability:**

3

---

### Official Review · Reviewer_CprD · 2024-05-20

**Rating:** 6
**Confidence:** 4

**Summary:**

The paper introduces a novel Fine-Grained Prompt Learning (FGPL) approach, leveraging Vision-Language Models (VLMs) like CLIP to address domain generalization challenges in Face Anti-Spoofing (FAS). By extracting both Domain-Agnostic (DA) and Domain-Specific (DS) features and employing an Adaptive Convolutional Adapter (AC-adapter), the method aims to improve the robustness and applicability of FAS systems across diverse datasets. Extensive experimental validation demonstrates the superiority of FGPL over state-of-the-art methods in cross-domain settings.

**Strengths:**

1. The paper effectively combines VLMs with prompt learning to extract and utilize DA and DS features, a method that stands out for its originality and potential to significantly enhance FAS domain generalization.
2. The experiments are thorough, with tests conducted across multiple challenging datasets, convincingly demonstrating that FGPL outperforms existing techniques. This robust experimental setup not only validates the effectiveness of the proposed method but also establishes a strong benchmark for future research.
3. The introduction of the AC-adapter is particularly noteworthy. This component integrates seamlessly with VLMs to bridge the gap between general image recognition tasks and FAS, providing a practical solution that enhances model adaptability without extensive retraining or manual tuning.

**Limitations:**

1. The paper brings textuality-based prompting information via CLIP into the process of FAS detection. It is hoped that subsequent experiments will be conducted to verify whether this leads to more information about the domain that cannot be generalised.
2. While the experimental results are strong, the paper could benefit from a deeper discussion on deployment challenges and limitations in real-world scenarios, including how the model performs under more realistic datasets.

**Suitability:**

3

---

### Official Review · Reviewer_6Ksj · 2024-05-24

**Rating:** 2
**Confidence:** 4

**Summary:**

This paper studies Domain-Generalized (DG) Face Anti-Spoofing (FAS), and introduces Fine-Grained Prompt Learning (FGPL), leveraging Vision-Language Models (VLMs) like CLIP. FGPL adaptively adjusts classifier weights using text features from prompts to mitigate classifier overfitting to different domain distributions. This approach uses prompts to capture both Domain-Agnostic and Domain-Specific features, enhancing category generalization through diversified Domain-Specific prompts. Additionally, we introduce an Adaptive Convolutional Adapter (AC-adapter) to bridge the gap between general image recognition and FAS tasks. Extensive experiments demonstrate the effectiveness of FGPL, outperforming existing methods across various cross-domain datasets.

**Strengths:**

1. The paper is well organized, the figures are readable and understandable.
2. The proposed method looks logical and technically sound.
3. The experimental results are good. Consistent improvements have been shown in different benchmarks.

**Limitations:**

1. Concerns on the novelty: Previous work  [0] has also explored fine-grained prompt learning for domain generalization, and STYLIP  [0] also fuses the domain-agnostic features and domain-specific features. Directly applying a similar idea to DG FAS seems does not introduce sufficient novelties. Besides, CFPL-FAS [1] also explored DG FAS via textual prompt learning. Please the authors make extensive discussions on the differences from these works.

[0]. STYLIP: Multi-Scale Style-Conditioned Prompt Learning for CLIP-based Domain Generalization, WACV 2023;

[1]. CFPL-FAS: Class Free Prompt Learning for Generalizable Face Anti-spoofing, CVPR 2024;

2. Lack of comparison results. 1) Since STYLIP [0], CFPL-FAS [1] and FLIP [2] are the most related work, however, there are no comparison results with these methods [0-2] in different benchmarks, and it is questionable the presented method's superiority and advantages. Does the presented method outperform them in Protocol 1 and Protocol 2? 2) There are only comparison results on 3 source domains to the unseen domain, what are the DG results of limited source domains (two domains) or single source domain to the unseen domain?

[2]. FLIP: Cross-domain Face Anti-spoofing with Language Guidance, ICCV 2023;

3. Lack of citations: some important references [3-9] that also study cross-domain FAS are missing.  The reviewer suggests the authors add these missing references and briefly review them in the related work.

[3]. Domain Invariant Vision Transformer Learning for Face Anti-Spoofing, WACV 2023;

[4]. Adaptive Mixture of Experts Learning for Generalizable Face Anti-Spoofing, ACM MM 2022;

[5]. Learning Meta Pattern for Face Anti-Spoofing, IEEE TIFS 2022;

[6]. Domain Generalization for Face Anti-Spoofing via Negative Data Augmentation, IEEE TIFS 2023;

[7]. Selective Domain-Invariant Feature Alignment Network for Face Anti-Spoofing, IEEE TIFS 2023;

[8]. Test-Time Domain Generalization for Face Anti-Spoofing, CVPR 2024;

[9]. Suppress and Rebalance: Towards Generalized Multi-Modal Face Anti-Spoofing, CVPR 2024;

4.  Insufficient experimental analysis: There's a lack of comprehensive experimental analysis, including TSNE feature visualization, visualization of source and target domains, class visualization, and the distribution of individual adapters. These analyses are crucial for understanding the model's behavior and performance, identifying areas for improvement, and assessing how the model performs in different environments.

5. In addition to the results in the manuscript, More realistic FAS protocols in [10-11] including both 2D and 3D attacks should be compared. For example, MCO to CelebA settings could be compared.

[10]. Towards Unsupervised Domain Generalization for Face Anti-Spoofing, ICCV 2023;

[11]. Causal intervention for generalizable face anti-spoofing, ICME 2022;

6. Concerns on the reproducibility： Since CFPL-FAS [1] does not open source their code, it is questionable on the reproducibility of the proposed method.  The implementation details are too short to reimplement by the readers who want to follow this work. Would the authors release their code to the community?

**Suitability:**

2

---

### Meta-Review · Area_Chair_zEmi · 2024-06-30

**Recommendation:** Accept (Oral)
**Confidence:** 3

**Metareview:**

While one reviewer still have serious concerns, most other reviewers lean to acceptance. Strongly urge authors to carefully address reviewers comments, e.g., some important experimental comparison results with the related works STYLIP, CFPL-FAS, and FLIP, are still missing, important comparison results on protocols of limited some domains are still absent, and experimental analyses of TSNE distribution analyses are lacking.